# Amino-Functionalized Fe_3_O_4_@SiO_2_ Core-Shell Magnetic Nanoparticles for Dye Adsorption

**DOI:** 10.3390/nano11092371

**Published:** 2021-09-12

**Authors:** Chun-Rong Lin, Oxana S. Ivanova, Dmitry A. Petrov, Alexey E. Sokolov, Ying-Zhen Chen, Marina A. Gerasimova, Sergey M. Zharkov, Yaw-Teng Tseng, Nicolay P. Shestakov, Irina S. Edelman

**Affiliations:** 1Department of Applied Physics, National Pingtung University, Pingtung City 90003, Taiwan; stp92065@gmail.com (Y.-Z.C.); chargedw@yahoo.com.tw (Y.-T.T.); 2Kirensky Institute of Physics, FRC KSC SB RAS, 660036 Krasnoyarsk, Russia; irbiz@iph.krasn.ru (D.A.P.); alexeys@iph.krasn.ru (A.E.S.); zharkov@iph.krasn.ru (S.M.Z.); nico@iph.krasn.ru (N.P.S.); ise@iph.krasn.ru (I.S.E.); 3Institute of Engineering Physics and Radio Electronics, Siberian Federal University, 660041 Krasnoyarsk, Russia; marina_2506@mail.ru

**Keywords:** Fe_3_O_4_@SiO_2_, core-shell nanoparticles, magnetic properties, water pollutions, fluorescence, adsorption

## Abstract

Fe_3_O_4_@SiO_2_ core-shell nanoparticles (NPs) were synthesized with the co-precipitation method and functionalized with NH_2_ amino-groups. The nanoparticles were characterized by X-ray, FT-IR spectroscopy, transmission electron microscopy, selected area electron diffraction, and vibrating sample magnetometry. The magnetic core of all the nanoparticles was shown to be nanocrystalline with the crystal parameters corresponding only to the Fe_3_O_4_ phase covered with a homogeneous amorphous silica (SiO_2_) shell of about 6 nm in thickness. The FT-IR spectra confirmed the appearance of chemical bonds at amino functionalization. The magnetic measurements revealed unusually high saturation magnetization of the initial Fe_3_O_4_ nanoparticles, which was presumably associated with the deviations in the Fe ion distribution between the tetrahedral and octahedral positions in the nanocrystals as compared to the bulk stoichiometric magnetite. The fluorescent spectrum of eosin Y-doped NPs dispersed in water solution was obtained and a red shift and line broadening (in comparison with the dye molecules being free in water) were revealed and explained. Most attention was paid to the adsorption properties of the nanoparticles with respect to three dyes: methylene blue, Congo red, and eosin Y. The kinetic data showed that the adsorption processes were associated with the pseudo-second order mechanism for all three dyes. The equilibrium data were more compatible with the Langmuir isotherm and the maximum adsorption capacity was reached for Congo red.

## 1. Introduction

Silica (SiO_2_) is frequently used as a support-material in core-shell structures; it does not only help nanoparticles (NPs) to become stable at certain conditions, but also provides them an opportunity to be easily modified with other functional groups and, additionally, to be environmentally compatible. Furthermore, silica coated magnetic NPs can be dispersed in water without adding other surfactants due to the negative charges on the silica shells. The fields of application of Fe_3_O_4_@SiO_2_ NPs functionalized with different surfactants, and more frequently with amines, are different. A number of authors have demonstrated their effective applications in biology and medicine [1,2], in catalysis [3,4], and, especially, as effective adsorbents for the removal of pollutants from wastewaters [5,6,7,8,9,10,11,12]. In ref. [1], the amino modifications of the silica surface of Fe_3_O_4_@SiO_2_ NPs reduced the detrimental interactions with cellular membranes and prolonged the blood circulation time after in vivo administration. In ref. [12], the magnetic core-shell Fe_3_O_4_@SiO_2_ NPs synthesized by the modified Stöber method and functionalized with amino and carboxyl groups were used as a nano-adsorbent for scandium ions from aqueous solutions. In Refs. [7,9], the synthesized Fe_3_O_4_@SiO_2_-NH_2_ nanocomposites were embedded into the polyether-sulfone membranes with different concentrations via the phase inversion method. Due to the nanocomposite adsorption properties, the significant enhancement in efficiency of the modified membranes for the removal of Cd(II) ions and methyl red dye was achieved. A new type of magnetic fluorescent nanocomposite (Fe_3_O_4_@SiO_2_-NH_2_/CQDs) was prepared by bonding of carbon quantum dots (CQDs) with Fe_3_O_4_@SiO_2_-NH_2_ nanocomposites through amine-carbonyl interactions and used as a fluorescent probe to detect Cu^2+^ [13]. The authors of ref. [11] concluded that the strategy of coating with silica and amino group functionalization of Fe_3_O_4_ nanoparticles is useful for increasing the adsorption capacity. In particular, they obtained adsorption capacities of 29.3 and 28.6 mg/g for light green and brilliant yellow anionic dyes, respectively, using Fe_3_O_4_@SiO_2_@NH_2_ NPs, while 12.1 and 9.4 mg/g only for Fe_3_O_4_ NPs. Since this line of research is rather new, and the properties and application possibilities of NPs strongly depend on the details of their synthesis, the search for optimal synthesis conditions and the study of the properties of the functionalized particles by various methods can be considered as an urgent task.

The present work is devoted to the study of the morphology, magnetic and adsorption properties of Fe_3_O_4_ NPs obtained by the co-precipitation method and then coated with amorphous silica to produce Fe_3_O_4_@SiO_2_ core-shell NPs, which was functionalized by an amine group to fabricate Fe_3_O_4_@SiO_2_-NH_2_ nanocomposites. The methylene blue (MB), Congo red (CR) and eosin Y (EY) were selected as the typical organic cation and anion pollutants to test the ability of the prepared composite for the adsorptive removal of organic pollutants from water. The study of the fluorescent properties of eosin Y-doped NPs was also carried out.

## 2. Materials and Methods

### 2.1. Synthesis Procedure

Ferrous sulfate heptahydrate (FeSO_4_·7H_2_O) (>99%) was obtained from Sigma-Aldrich, ethanol (CH_3_CH_2_OH) (>95%) was obtained from Fullin Nihon Shiyaku Biochemical Ltd., tetraethyl orthosilicate (Si(OC_2_H_5_)_4_, TEOS) (>99%) and ammonia solution (28–30 wt.% NH_3_) were purchased from Acros Organics. Potassium nitrate (KNO_3_) (>99%) was purchased from Wako Pure Chemical Industries, Ltd. Sodium hydroxide pellets was obtained from PanReac AppliChem. Poly(sodium 4-styrene-sulfonate) solution (30 wt.%) in water was purchased from Aldrich. All the chemicals were used without further purification. Several stages were used to synthesize nanocomposites.

Magnetic NPs, Fe_3_O_4_, were produced by co-precipitation, hydrolysis of iron sulfate under Ar flow. In the typical synthesis, a mixture of the basic solution is 0.02 mol KNO_3_ and 0.2 mol NaOH in 50 mL of ultrapure water, previously deoxygenated, was prepared under Ar and, then was added dropwise to the Fe (II) solution (one is 0.25 M, the other is 0.75 M, and mixed together), under Ar and at 90 °C. The black precipitate formed was stirred over 1 h at 90 °C, then, washed thoroughly with ethanol. Silica coating of the magnetite nanoparticles was performed via the Stöber process. First, Fe_3_O_4_ NPs (200 mg) were dispersed in ethanol (150 mL) and kept immersed in a water bath over 15 min under sonication. Then, an ammonia solution (6 mL) and TEOS (200 μL) were slowly added to the Fe_3_O_4_ suspension. Finally, the SiO_2_ coated particles were collected magnetically using an NdFeB magnet, and the isolated powders were washed thoroughly with ethanol (sample Fe_3_O_4_@SiO_2_). The amino functionalization of silica coated NPs was made in the same manner as is mentioned above, replacing TEOS to (3-aminopropyl)-triethoxysilane (APTES) (sample Fe_3_O_4_@SiO_2_-NH_2_). At last, eosin Y was attached to the amino silanized magnetic nanoparticles covalently. First, Fe_3_O4@SiO_2_-NH_2_ NPs (200 mg) were dispersed in ultrapure water (50 mL) and over 15 min were exposed to sonication. The 0.2 mmol eosin Y and catalyst were added to the NPs suspension under stirring during 1 h, then, washed thoroughly with ethanol (sample Fe_3_O_4_@SiO_2_-NH_2_-EY).

### 2.2. Characteristic Methods

The crystal structure of the NPs was characterized by X-ray diffraction measurements using a Bruker D8 Advance diffractometer (Cu Kα radiation, 40 kV, 25 mA, λ = 1.5418 Å) (Bruker Optik GmbH, Ettlingen, Germany).

The morphology and microstructure of the NPs were investigated using transmission electron microscopy (TEM). TEM experiments were performed with a JEM-2100 (JEOL Ltd., Tokyo, Japan) microscope operating at the accelerating voltage of 200 kV (Siberian Federal University) and JEOL JEM-1230 microscope (JEOL Ltd., Tokyo, Japan) operated at an accelerating voltage of 80 kV (Precision Instruments Center of NPUST). Selected-area electron diffraction (SAED) was used to determine the structure of the NPs.

Fourier transform infrared absorption (FT-IR) spectra were recorded with a VERTEX 70 (Bruker Optik GmbH, Ettlingen, Germany) spectrometer in the spectral region of 400 ÷ 4000 cm^−1^ with the resolution 4 cm^−1^. The spectrometer was equipped with a Globar as the light source and a wide band KBr beam splitter and RT-DLaTG as the detector (Bruker Optik GmbH). For the measurements, round tablet samples of about 0.5 mm thick and of 13 mm in diameter containing NPs were prepared as follows: nanopowders in amount lower than 0.001 g were thoroughly ground with 0.14 g of KBr; the mixtures were formed into tablets which were subjected to cold pressing at 10,000 kg.

The magnetic properties were measured with the vibrating sample magnetometer Lakeshore 7400 series VSM (Lake Shore Cryotronics, Inc., Westerville, OH, USA).

The excitation and fluorescence spectra were measured on a Fluorolog 3–22 spectrofluorometer (Horiba Jobin Yvon, Edison, NJ, USA). The obtained spectra were corrected for sensitivity of PMT, reabsorption effects and background. The quartz cells with the cross sections of 10 × 10 mm^2^ were used to investigate the solutions for L-geometry of excitation. All the measurements were carried out at room temperature.

The absorption spectra were recorded with a UV/VIS circular dichroism spectrometer SKD-2MUF (OEP ISAN, Moscow, Russia). Quartz cells with the optical path length of 5 mm were used.

## 3. Results and Discursion

### 3.1. NPs Structure and Morphology

The XRD patterns (Figure 1) revealed that the parent NPs and magnetic cores of all the nanocomposites were of the spinel ferrite crystal structure with the parameters of the most intense peaks corresponding to the Fe_3_O_4_ phase (PDF Card # 04-005-4319).

The TEM images (Figure 2a–d) revealed NPs of predominantly rectangular shape with the average size of 25 ± 5 nm (Figure 2e). At the same time, a small amount of ellipsoidal NPs with the sizes of about 10 nm and rectangular NPs of a larger size can be noticed. The almost ideal crystal structure of the initial Fe_3_O_4_ NPs is seen very well in the HRTEM image in Figure 3a. As a result of the silica coating, the initial NPs became covered with a homogeneous amorphous silica shell, 6–7 nm thick seen especially well in the HRTEM image (Figure 3b). Functionalization with NH2 and further doping with eosin Y led to some blurring of the HRTEM image (Figure 3c,d). The selected area electron diffraction (SAED) patterns also shown in Figure 3c,d confirm the presence of the Fe_3_O_4_ crystalline core in all NPs.

The FT-IR spectra (Figure 4) show the appearance of new bands upon the transition Fe_3_O_4_ → Fe_3_O_4_@SiO_2_ → Fe_3_O_4_@SiO_2_-NH_2_ → Fe_3_O_4_@SiO_2_-NH_2_-EY evidencing the appearance of chemical bonds. In the spectrum of pure Fe_3_O_4_ NPs, the strong band at 580 cm^−1^ is due to Fe-O stretching vibrations in accordance with other authors [9,11,13]. This band is observed in the spectra of all the samples. The wide asymmetric band at 1094 cm^−1^ appeared in the spectrum of the Fe_3_O_4_@SiO_2_ sample and is seen in the spectra of two next samples having a SiO_2_ shell. This band can be related to the asymmetric stretching vibrations of Si-O-Si. An analogous band was observed and interpreted equally by all the authors studying Fe_3_O_4_@SiO_2_ NPs [9,11,13]. Symmetric Si-O-Si stretching vibrations were associated in [9] with the weak band near 808 cm^−1^. The same weak band is seen at 800 cm^−1^ in Figure 4. At the same time, the authors of ref. [13] associated the band at 797 cm^−1^ with the Si-O-Fe stretching vibrations proving the presence of a chemical bond between the magnetic core and silica shell [9]. Only minor changes appear in the spectra of the Fe_3_O_4_@SiO_2_-NH_2_ and Fe_3_O_4_@SiO_2_-NH_2_-EY samples. The appearance of a wide band near 3430 cm^−1^ is a more noticeable feature characteristic of the NH_2_ amino group [14]. A small amount of covalently attached eosin Y in comparison with Fe_3_O_4_@SiO_2_-NH_2_ NPs does not allow one to clearly see changes in the bending and stretching vibrations of the amino groups.

### 3.2. NPs Magnetic Properties

The magnetic measurements (Figure 5) show very narrow hysteresis loops with magnetic saturation in the external magnetic field near 3 kOe and coercive force of about 100 Oe. The saturation magnetization Ms value of the initial Fe_3_O_4_ NPs is exceptionally high, significantly higher than the one presented by other authors for Fe_3_O_4_ NPs, for example [11] and even higher than in bulk Fe_3_O_4_ samples, 92 emu/g at room temperature [15]. The redistribution of Fe ions between the oppositely magnetized tetrahedral and octahedral sublattices in magnetite NPs caused, for example, by the technological conditions can be one of the reasons of the observed Ms increase. Since the resulting magnetization of the sample is due to the difference between the magnetic moments of Fe ions occupying octahedral and tetrahedral positions, the Fe ion deficiency in tetrahedral position can be responsible for an increase of the sample magnetization.

The surface modifications lead to the Ms decrease. However, it remains quite high. A similar decrease in the saturation magnetization of Fe_3_O_4_ NPs coated with silica compared to the initial Fe_3_O_4_ nanoparticles was noted by a number of authors. For example, in ref. [11] Ms was equal to 38, 23, and 11 emu/g for Fe_3_O_4_, Fe_3_O_4_@SiO_2_ and Fe_3_O_4_@SiO_2_@NH_2_ NPs, correspondingly; in ref. [16]—80, 31 and 20 emu/g for Fe_3_O_4_, Fe_3_O_4_@SiO_2_ and Fe_3_O_4_@SiO_2_-NH_2_ NPs respectively.

The decrease in magnetization of NPs after coating with silicon oxide(curve 2 in Figure 5) can be caused by a number of factors. First of all, when determining the magnetization value, the whole particle mass was taken into account including the silica shell. In addition, the shell can affect the spin state of the magnetic core surface layers. However, the addition of the amino groups and then eosin Y resulted in an increase in magnetization. Such a behavior of the magnetization was not previously observed in the above-cited and other works. It is possible that, in the process of amino functionalization, cation redistribution between the sublattices continued. This question needs special consideration.

### 3.3. Application of Synthesized NPs as Fluorescent Probes

The development of hybrid nanoparticle technology, synthesized fluorescent nanoparticles with encapsulated quantum dots [17], or dyes [18] have attracted great interest in recent years. Despite the excellent brightness and photostability of quantum dots for imaging applications, the risk of systemic toxicity remains high due to the incorporation of heavy metals. Thus, dye-doped nanoparticles still appear very promising. The simultaneous combination of fluorescent and magnetic properties of NP imaging would greatly benefit in the diagnostics and monitoring of living cells and organisms [19,20].

The spectral properties of the synthesized EY-doped Fe_3_O_4_@SiO_2_@NH_2_ dispersed in water solution are presented in Figure 6a. The fluorescence spectrum turned out to be independent of the excitation wavelength and NPs give the green emission with the maximum at 542 nm with the excitation maximum at 515 nm. The fluorescent spectrum of EY-doped NPs dispersed in solution displays a red shift of 6 nm and 40 % broadening as compared with the dye molecules free in water in Figure 6b. On the contrary, the maximum of the excitation spectrum (measured at 580 nm) shows a slight blue shift of 1 nm and 30% broadening. The reasons for the observed spectral changes could include the change in the ionic form of the EY molecules and polarity decrease in the microenvironment. In the water solution, pH 6, most of the EY molecules are in the dianionic form. When attached to the amino groups in NPs, the dye changes its ionic form to the anionic or even neutral one and the spectral properties of these forms are different, on the one hand [21]. On other hand, as the solvent polarity decreases, the red shift in the emission of EY is observed [22].

### 3.4. Application of Synthesized NPs for Dye Adsorption

#### 3.4.1. Adsorption Kinetics

The spectral properties of two anionic (eosin Y (EY) and Congo red (CR), and one cationic) methylene blue (MB) dyes were used to find out the adsorption capacity of amino-functionalized Fe_3_O_4_@SiO_2_ core-shell magnetic NPs in distilled water (measured pH 5.5) at 25 °C. The dye concentration was determined by absorbance at the wavelength corresponding to the maxima in the spectra of 490 nm for eosin Y, 505 nm for CR, and 664 nm for MB.

For a typical experiment shown schematically in Figure 7, 3 mg of adsorbent was dispersed in 1.5 mL of the dye aqueous solution at the initial concentration of the dye *C*_0_ = 30 mg/L. The solution was placed in an ultrasonic bath for 10 min for intensive mixing. Then, a magnetic nanoadsorbent was separated from the solution by applying a magnetic field and the absorption spectra of the solution were measured. The shaking and magnetic separation was repeated multiple times to obtain kinetic curves. The value of the adsorption capacity *q_t_* (mg/g) of NPs was calculated as follows
(1)$$q_{t}=\frac{(C_{0}-C_{t})V}{m},$$
where *C*_0_ and *C_t_* (mg/L) are the initial concentration and concentration of the dye at the contact time *t*, *V* (L) is the volume of the solution; and *m* (g) is the mass of the adsorbent (NPs).

The effect of the contact time on the adsorption of Fe_3_O_4_@SiO_2_-NH_2_ NPs is shown in Figure 8 for the dyes at the same initial concentration of *C*_0_ = 30 mg/L. At the initial stage of adsorption kinetics, the amount of the adsorbed dye onto the magnetic NPs increases rapidly due to a large number of vacant active sites on the surface of the amino-functionalized silica coated magnetic NPs. After that, these surface sites are gradually occupied, so the adsorption rate decreases until the adsorption equilibrium is established. The time to reach the equilibrium adsorption for the anionic dyes on Fe_3_O_4_@SiO_2_-NH_2_ NPs is much faster than for the cationic dye: 60 min for EY, 100 min for CR, and more than 300 min for MB. The other adsorbent Fe_3_O_4_@SiO_2_@NH_2_@Zn−TDPA NPs [TDPAT = 2,4,6-tris(3,5-dicarboxyl phenylamino)-1,3,5-triazine] showed the similar time of 120 min both for the cationic MB and anionic CR dye [23]. The adsorption of CR by the Fe_3_O_4_@SiO_2_ nanospheres achieved the equilibrium only after 5 h [24]. The equilibrium adsorption for initial concentration of *C*_0_ = 30 mg/L was measured after 24 h of contact of NPs with the dye solution. The equilibrium adsorption capacity qe of CR (11.6 mg/g) is higher than that of EY (7.2 mg/g) and MB (9.8 mg/g).

A quantitative understanding of the adsorption is possible with the help of kinetic models. The adsorption kinetics of the dyes on the magnetic NPs was described by the kinetic models of the pseudo-first order:(2)$$\ln\left(q_{e}-q_{t}\right)=\ln q_{e}-k_{1}t$$
and of the pseudo-second order:(3)$$\frac{t}{q_{t}}=\frac{1}{k_{2}\;q_{e}^{2}}+\frac{t}{q_{e}},$$
where *q_e_* and *q_t_* (mg/g) are the amounts of the dye at equilibrium and at contact time *t,* respectively; *k*_1_ (1/min) and *k_2_* (g/(mg min)) are the adsorption rate constants of the reaction of the pseudo-first and pseudo-second orders, respectively.

The curve-fitting plots of the adsorption ability of the dyes by Fe_3_O_4_@SiO_2_-NH_2_ NPs using the two kinetic models are shown in Figure 9, and the fitted parameters with the correlation coefficients *R*^2^ are summarized in Table 1. The *R*^2^ values for the pseudo-second order kinetic model (*R*^2^ = 0.995–0.999) were higher than those of the pseudo-first order model (*R*^2^ = 0.903–0.970). In addition, the calculated values of *q*_e_ (Table 1) determined by the pseudo-second order model are more consistent with the measured values of *q*_e_ than that of the pseudo-first-order model. These results prove that the adsorption process of these dyes on Fe_3_O_4_@SiO_2_-NH_2_ NPs completely followed a the pseudo-second order kinetic model, suggesting that adsorption is dependent on the amount of the solute adsorbed on the surface of the adsorbent and the amount of active sites. It should be noted that the dye adsorption kinetics on magnetic nanoparticles is most often described in terms of the pseudo-second order model [23,24,25,26,27,28,29,30].

The intraparticle diffusion model is often used to identify diffusion mechanisms [23,25,26,27,28,30,31]. In this model, the rate of intraparticle diffusion is a function of *t*^0.5^ and can be determined as follows:(4)$$q_{t}=k_{i}t^{0.5}+C,$$
where *k*_i_ is the intraparticle diffusion rate constant, (mg/g.min^0.5^). *C* is the intercept of the linear curve. According to the model proposed by Weber and Morris [32], the adsorption process is controlled only by intraparticle diffusion if the plot is a straight line and passes through the origin. Otherwise, if the plot is multilinear or does not pass the origin, more than one diffusion mechanism might determine the adsorption process and adsorption is related to diffusion within the particles.

As shown in Figure 10, at least two stages are observed for anionic CR, EY as well as for the cationic MB dye, and thus, two or more diffusion mechanisms can affect the adsorption. The first stage refers to the transport of the dye molecules from the solution to the external surface of NPs. This stage is completed after up to 60 min for EY, 90 min for CR, and 240 min for MB. The second stage corresponds to the diffusion of the dye molecules within the micropores of Fe_3_O_4_@SiO_2_-NH_2_ NPs. The high initial absorption rates k_i1_ of the first stage (Table 1) are observed for all the dyes, indicating a fast initial dye removal process and the predominant role of external surface diffusion, especially, for CR. Extremely low adsorption rates *k_i_*_2_ (8–25 fold less than for the first stage) shows a negligible proportion of intraparticle diffusion of the dye molecules within the micropores of NPs. A similar two-stage adsorption process was observed for MB onto B-Fe_3_O_4_@C NPs [28], for CR and MB dyes onto Fe_3_O_4_@SiO_2_@NH_2_@Zn−TDPA NPs [23], and for EY onto Fe_3_O_4_/polypyrrole composites [30].

#### 3.4.2. Adsorption Isotherms 

Two adsorption isotherm models (Langmuir and Freundlich) were applied to understand the adsorbate–adsorbent interaction. The Langmuir equation was defined as follows
(5)$$q_{e}=\frac{\;q_{\max}\;K_{L}C_{e}}{K_{L}C_{e}+1},\;$$
where *q_e_* (mg/g) is the amount of the dye adsorbed at the equilibrium, *q*_max_ represents the maximum adsorption capacity (mg/g), *K_L_* is the Langmuir adsorption constant (L/mg), and *C_e_* is the equilibrium concentration of the adsorbed dye (mg/L).

The Freundlich equation is expressed as:(6)$$q_{e}=K_{F}C_{e}^{1/n},$$
where *K_F_* is the Freundlich adsorption constant (L/mg); the dimensionless constant 1/*n* is an empirical parameter related to the isotherm shape. Based on the 1/*n* values, the adsorption process can be classified as irreversible (1/*n* = 0), favorable (0 < 1/*n* < 1), or unfavorable (1/*n* > 1) [33].

The Langmuir model illustrates the formation of a homogeneous adsorbed monolayer, while the adsorbed molecules do not interact with each other. The Freundlich model considers the existence of a more complicated multilayered structure.

Based on the results presented above, we have chosen the anionic CR dye for further investigation owing to its highest capacity and shortest time of the adsorption process. The adsorption isotherm models were used to describe the equilibrium between the adsorbed CR on the surface of Fe_3_O_4_@SiO_2_-NH_2_ NPs (Figure 11). The obtained parameters are presented in Table 2. According to the correlation coefficients, the Langmuir model (*R*^2^ = 0.967) coincided with the experimental data much better than the Freundlich model (*R*^2^ = 0.789), which indicates that the homogeneous and monolayer adsorption is the dominant adsorption in the case of CR. Although the 1/*n* value is lower than 1 and indicates that the adsorption of CR is favorable, the *R^2^* values of the Freundlich isotherm do not fall within the acceptable range. In contrast to our results, the adsorption isotherm of CR on Fe_3_O_4_@SiO_2_@Zn-TDPAT NPs was satisfactorily described both by the Langmuir and Freundlich models [23].

The adsorbents based on Fe_3_O_4_@SiO_2_ nanoparticles for the removal of the anionic CR and cationic MB dyes are summarized in Table 3. The maximum adsorption capacity *q*_max_ of the synthesized amino-functionalized Fe_3_O_4_@SiO_2_ core-shell magnetic nanoparticles is comparable to other analogous adsorbents [10,23,24,29,34].

As is seen from Table 3, the adsorption efficiency can further be improved by the surface modification of NPs. It should be emphasized that the pH and ionic strength of the aqueous solution [23] as well as the temperature [24] could also significantly influence the efficiency of the dye adsorption on NPs due to the involvement of different adsorption mechanisms including electrostatic and hydrophobic interactions, hydrogen bonding, van der Waals forces, etc. The measured zeta potential of the Fe_3_O_4_@mSiO_2_-NH_2_ nanoparticles vs. pH showed [35] that the isoelectric point was at 5.9 due to the amino-functionalization, denoting that the modified nanoparticles would be positively charged at pH < 5.9 and negatively charged at pH > 5.9. Under the conditions of our experiment (the initial pH 5.5), the surface of the magnetic NPs had a slightly positive charge. Table 3 indicates that the maximum adsorption capacity *q*_max_ is close for both anionic CR and cationic MB dyes, so the electrostatic interactions are not only responsible for the adsorption of the dyes on NPs.

## 4. Conclusions

In this study, core-shell magnetic nanoparticles, Fe_3_O_4_@SiO_2_, were synthesized by silica coating of the initial Fe_3_O_4_ NPs via the Stöber process and then functionalized with amino groups NH_2_. X-ray and electron diffraction data showed the magnetic core of the particles to have the magnetite Fe_3_O_4_ crystal structure without the presence of any other phases. Transmission electron microscopy showed predominantly rectangular NPs with the average size of 25 ± 5 nm in the initial Fe_3_O_4_ powder sample. The homogeneous amorphous silica shells with the thickness of about 7 nm were formed around each initial NP. The FT-IR spectra confirmed the appearance of the chemical bonds between the silica shell and magnetic core of NPs as well between the silica and amino group. The magnetic measurements revealed unusually high saturation magnetization Ms of the initial Fe_3_O_4_ NPs even higher than this value of the bulk magnetite crystal. Ms of the functionalized samples also significantly exceeded Ms of similar samples presented in literature. The high Ms value can be considered as an advantage of the studied nanomaterials since higher magnetization requires the use of weaker magnetic fields to control the processes involving these materials.

The fluorescence spectrum of EY-doped Fe_3_O_4_@SiO_2_@NH_2_ NPs dispersed in water solution was studied. NPs gave the green emission with the maximum at 542 nm with the excitation maximum at 515 nm. The spectrum displayed a red shift of about 6 nm and 40% broadening as compared to the dye molecules free in water while the maximum of the excitation spectrum (measured at 580 nm) showed a slight blue shift of 1 nm and 30% broadening. The observed spectral changes were associated with the change in the ionic form of the EY molecules due to the attachment to the amino groups in NPs.

The dye adsorption capacity and kinetics of the amino-functionalized Fe_3_O_4_@SiO_2_ core-shell magnetic NPs were studied in application to two anionic (eosin Y (EY) and Congo red (CR), and one cationic) methylene blue (MB) dyes. It was shown that the adsorption process of these dyes on the studied NPs followed the pseudo-second order kinetic model, suggesting that sorption is dependent on the amount of the solute adsorbed on the surface of the adsorbent and the amount of active sites. At least two stages were revealed in the adsorption time dependence for all three dyes. In the first stage, the transport of the dye molecules from the solution to the external surface of NPs occurs. The second stage corresponds to the diffusion of the dye molecules within the micropores of NPs. The high initial absorption rates of the first stage were observed for all the dyes, indicating a fast initial dye removal process and the predominant role of the external surface diffusion, especially, for CR. It was shown that the experimental data were fitted to the Langmuir model of the adsorption processes indicating that the homogeneous and monolayer adsorption was the dominant adsorption in the considered cases. The maximum adsorption capacity of the synthesized amino-functionalized Fe_3_O_4_@SiO_2_ core-shell magnetic nanoparticles is comparable to other analogous adsorbents presented in literature.

## Figures and Tables

**Figure 1 nanomaterials-11-02371-f001:**
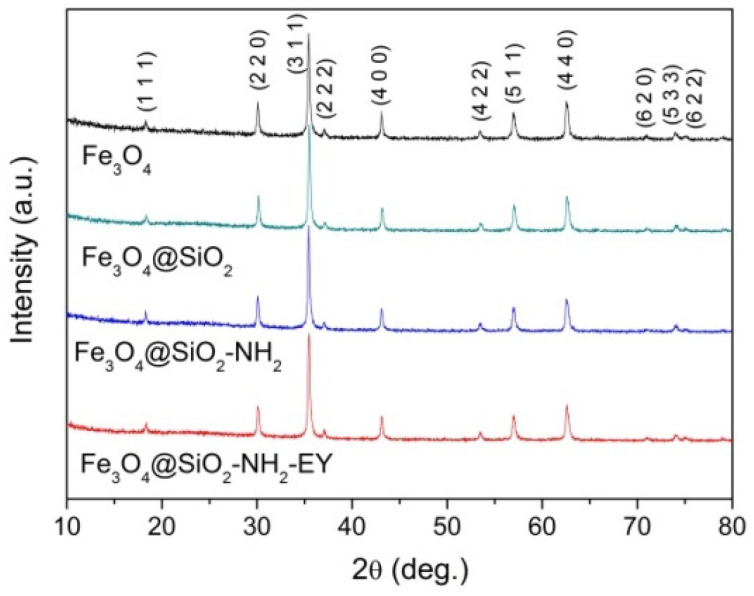
XRD patterns of synthesized NPs. The reflexes corresponding to PDF Card # 04-005-4319 are indicated in parenthesis.

**Figure 2 nanomaterials-11-02371-f002:**
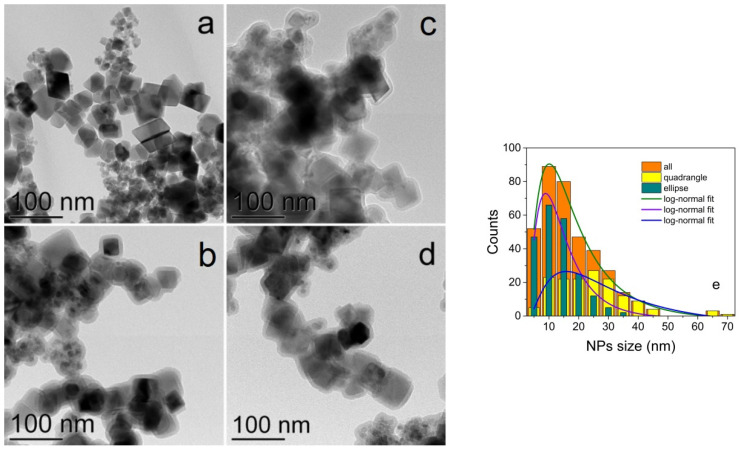
TEM images of Fe_3_O_4_ (**a**), Fe_3_O_4_@SiO_2_ (**b**), Fe_3_O_4_@SiO_2_-NH_2_ (**c**) and Fe_3_O_4_@SiO_2_-NH_2_-EY (**d**) NPs. Size distribution of Fe_3_O_4_ NPs (**e**).

**Figure 3 nanomaterials-11-02371-f003:**
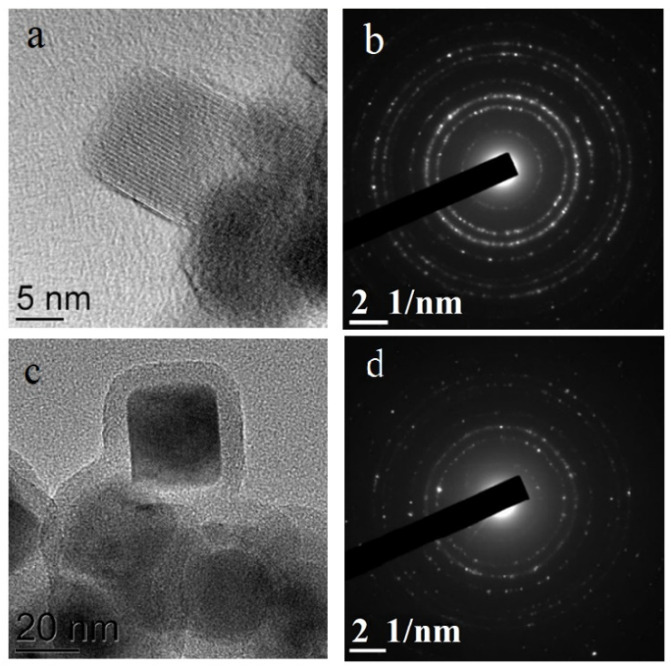
HRTEM image and selected area electron diffraction (SAED) of Fe_3_O_4_ (**a**,**b**) and Fe_3_O_4_@SiO_2_-NH_2_ (**c**,**d**) NPs, respectively.

**Figure 4 nanomaterials-11-02371-f004:**
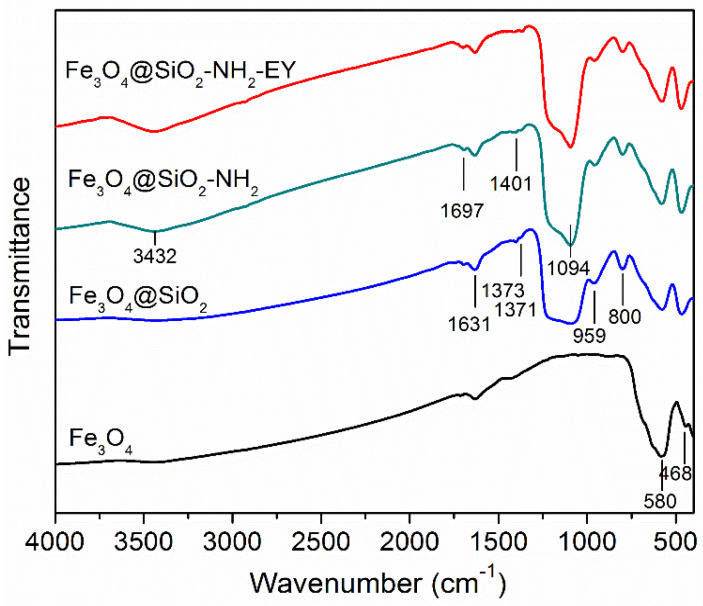
FT-IR spectra for Fe_3_O_4_, Fe_3_O_4_@SiO_2_, Fe_3_O_4_@SiO_2_-NH_2_, and Fe_3_O_4_@SiO_2_-NH_2_-EY.

**Figure 5 nanomaterials-11-02371-f005:**
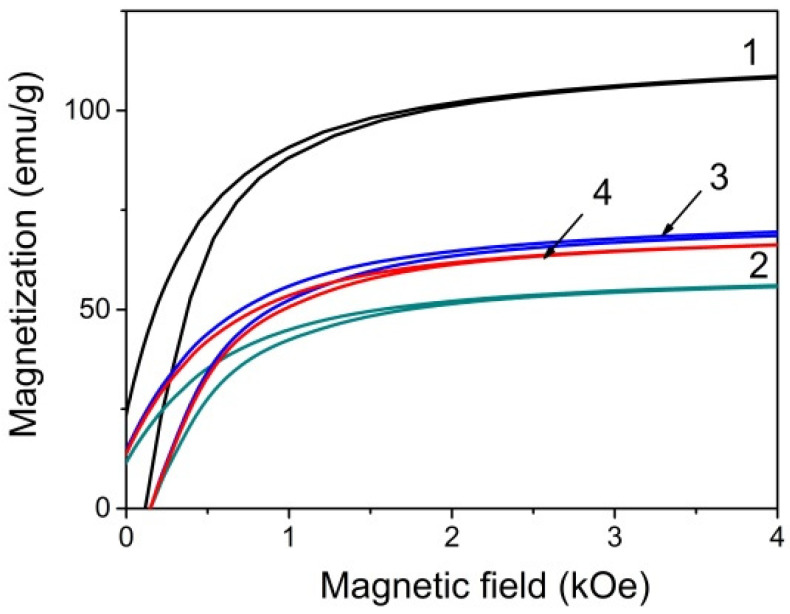
Room temperature magnetization curves for all NPs: Fe_3_O_4_ (1), Fe_3_O_4_@SiO_2_ (2), Fe_3_O_4_@SiO_2_-NH_2_ (3), and Fe_3_O_4_@SiO_2_-NH_2_-EY (4).

**Figure 6 nanomaterials-11-02371-f006:**
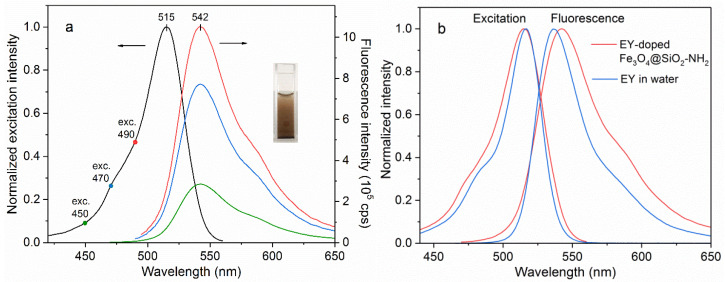
(**a**) Excitation spectrum and fluorescence spectrum of EY-doped Fe_3_O_4_@SiO_2_@NH_2_ dispersed in water under different excitation wavelengths. (**b**) Spectral changes of EY-doped NPs in comparison with EY free in distilled water (measured pH 5.5).

**Figure 7 nanomaterials-11-02371-f007:**
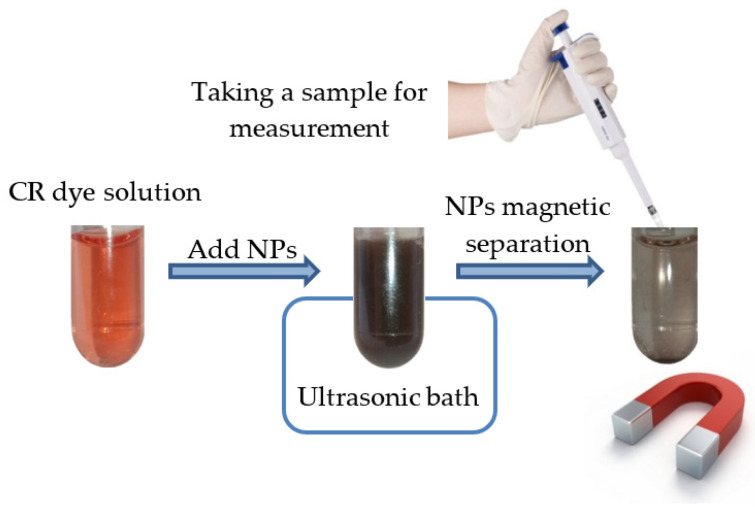
Schematic picture showing the preparation of a sample to measure the adsorption capacity, using the CR (*C*_0_ = 30 mg/L) water solution as an example.

**Figure 8 nanomaterials-11-02371-f008:**
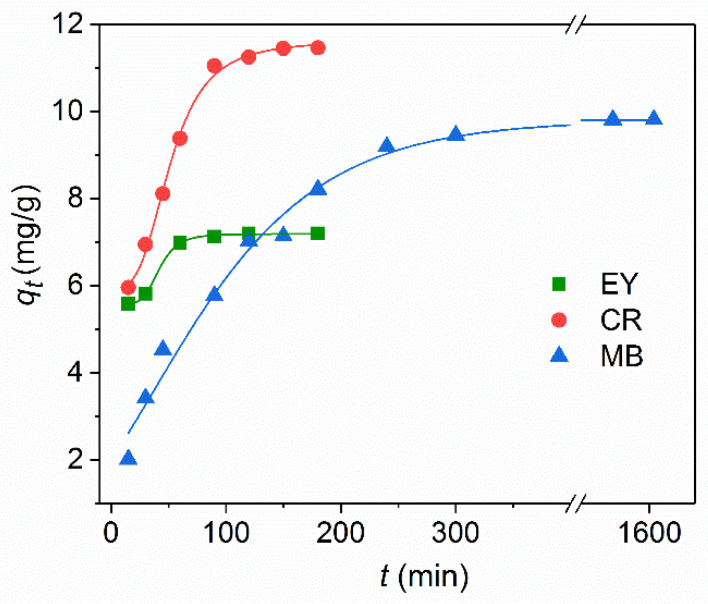
The effect of contact time on the dye adsorption for Fe_3_O_4_@SiO_2_@NH_2_ NPs at 25 °C. Experimental conditions: *C*_0_ = 30 mg/L, *m*(NPs) = 3 mg in *V* = 1.5 mL.

**Figure 9 nanomaterials-11-02371-f009:**
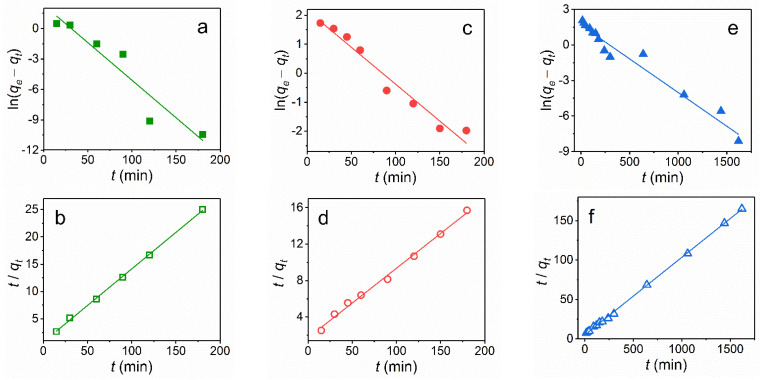
The pseudo-first-order (**a**,**c**,**e**) and pseudo-second-order (**b**,**d**,**f**) kinetics of EY (**a**,**b**), CR (**c**,**d**) and MB (**e**,**f**) adsorption on amino-functionalized Fe_3_O_4_@SiO_2_ core-shell magnetic NPs at 25 °C.

**Figure 10 nanomaterials-11-02371-f010:**
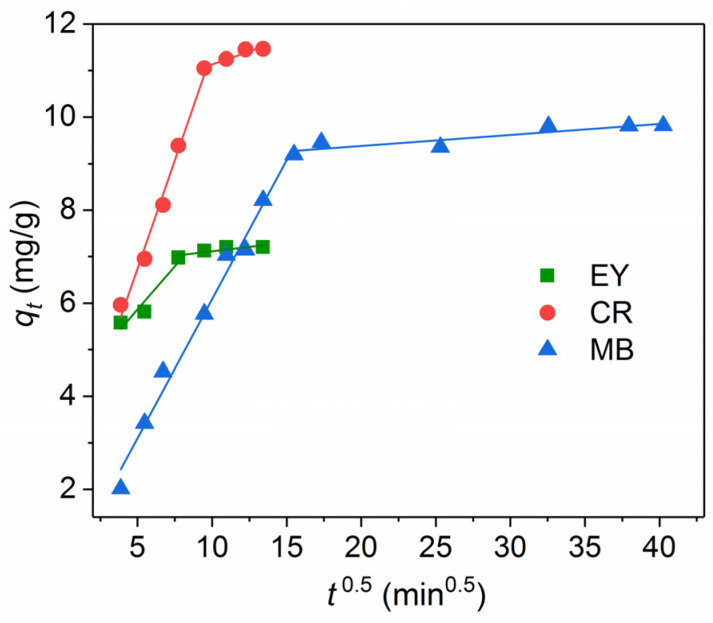
Intraparticle diffusion kinetic model of the dye adsorption on the amino-functionalized Fe_3_O_4_@SiO_2_ core-shell magnetic NPs at 25 °C.

**Figure 11 nanomaterials-11-02371-f011:**
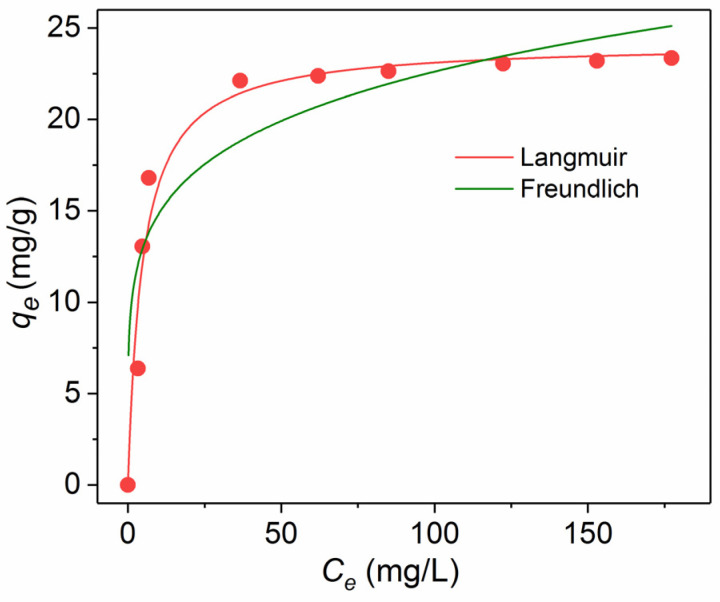
Adsorption isotherms of CR on magnetic NPs at 25 °C.

**Table 1 nanomaterials-11-02371-t001:** Kinetic parameters (Equations (2)–(4)) for the adsorption of the dyes (initial *C*_0_ = 30 mg/L) on Fe_3_O_4_@SiO_2_-NH_2_ NPs at 25 °C.

Kinetics	Parameters	EY	CR	MB
Pseudo-first order model	*k*_1_ (1/min)	0.074 ± 0.012	0.025 ± 0.002	0.0057 ± 0.0003
	*q_e_* (mg/g)	10.2 ± 3.4	8.7 ± 1.4	5.3 ± 1.2
	*R* ^2^	0.903	0.964	0.970
Pseudo-second order model	*k*_2_ (g/(mg min))	0.023 ± 0.005	0.0034 ± 0.0005	0.0019 ± 0.0002
	*q_e_* (mg/g)	7.5 ± 0.6	12.6 ± 0.9	10.2 ± 0.8
	*R* ^2^	0.999	0.995	0.999
Intraparticle diffusion model	*k_i_*_1_ (mg/(g min^0.5^))	0.37 ± 0.07	0.93 ± 0.06	0.60 ± 0.03
	*C*_1_ (mg/g)	4.0 ± 0.6	2.1 ± 0.3	0.11 ± 0.03
	*R* ^2^	0.938	0.988	0.986
	*k_i_*_2_ (mg/g min^0.5^)	0.04 ± 0.01	0.11 ± 0.02	0.024 ± 0.006
	*C*_2_ (mg/g)	6.7 ± 0.2	10.0 ± 0.3	8.9 ± 0.2
	*R* ^2^	0.873	0.929	0.902

**Table 2 nanomaterials-11-02371-t002:** Adsorption isotherm parameters (Equations (5) and (6)) for Congo red at 25 °C.

Langmuir Model			Freundlich Model		
*q*_max_, mg/g	*K_L_*, L/mg	*R* ^2^	1/*n*	*K_F_*, L/mg	*R* ^2^
24 ± 1	0.21 ± 0.04	0.967	0.18 ± 0.04	0.010 ± 0.002	0.789

**Table 3 nanomaterials-11-02371-t003:** Comparison of the adsorption capacity of the Fe_3_O_4_@SiO_2_ core-shell magnetic nanoparticles for the anionic CR and cationic MB dyes.

Anionic Dye	Adsorbent	*q*_max_ (mg/g)	Cationic Dye	Adsorbent	*q*_max_ (mg/g)
CR	Fe_3_O_4_@SiO_2_-NH_2_	24.0 [This work]	MB	Fe_3_O_4_@SiO_2_-NH_2_	20 *^a^* [This work]
	Fe_3_O_4_@SiO_2_-NH_2_	18.0 [29]		Fe_3_O_4_@SiO_2_	31.8 [24]
	Fe_3_O_4_@SiO_2_-NH_2_-MIP *^b^*	35.3 [29]		Fe_3_O_4_@SiO_2_	32.3 [23]
	Fe_3_O_4_@SiO_2_	36.2 [24]		Fe_3_O_4_@SiO_2_-CR *^c^*	31.4 [34]
	Fe_3_O_4_@SiO_2_	14.8 [23]		Fe_3_O_4_@SiO_2_@Zn−TDPAT *^d^*	58.7 [23]
	Fe_3_O_4_@SiO_2_@Zn-TDPAT *^d^*	17.7 [23]		Fe_3_O_4_@SiO_2_-(CH_2_)_3_-IL/Talc *^e^*	6.2 [10]

*^a^ q*_max_ = *q_e_* at *C*_0_ = 60 mg/L. ^*b*^ MIP is the molecularly imprinted polymer. *^c^* CR is Congo red. *^d^* TDPAT is 2,4,6-tris(3,5-dicarboxyl phenylamino)-1,3,5-triazine. ^*e*^ IL is the ionic liquid.

## Data Availability

The data presented in this study are available on request from the corresponding author. The data are not publicly available since they are a part of ongoing research.

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
