# Peer review of "Amino-Functionalized Fe3O4@SiO2 Core-Shell Magnetic Nanoparticles for Dye Adsorption"

_nanomaterials, 2021, doi:10.3390/nano11092371_

Round 1

Reviewer 1 Report

REPORT

“Amino-functionalized Fe3O4@SiO2 core-shell magnetic nano-particles for dye adsorption”

The authors have prepared Fe3O4@SiO2 core-shell nanoparticles functionalized with NH2 amino-groups.The samples were characterized by X-ray diffraction, FT-IR spectroscopy, transmission electron microscopy, selected area electron diffraction and vibrating sample magnetometry. The fluorescent spectrum of eosin Y - EY-doped sample were measured and analyzed

The results are very are very interesting and consistent, with a good discussion. It is well referenced and the references are updated.

Only some small remarks:Abstract, line 23 – What is “EY-doped NPs”? I suggest that the first time an acronym appears it should be defined.Page 2, line – 53, instead of Fe3O4 it should be Fe3O4, and line 354.Page9, line 285 – sorption?? Adsorption??

In my opinion the article can be accepted for publication in the journal “Nanomaterials” after the suggested corrections are made.

Author Response

Remark 1. Abstract, line 23 – What is “EY-doped NPs”? I suggest that the first time an acronym appears it should be defined.

Answer. Acronym is defined.

Remark 2. Page 2, line – 53, instead of Fe3O4 it should be Fe3O4, and line 354.

Answer. Fe3O4 is changed for Fe3O4

Remark 3. Page9, line 285 – sorption?? Adsorption??

Answer. The term is changed.

Reviewer 2 Report

The present manuscript reports the investigation of Fe3O4@SiO2 core-shell nanoparticles functionalized with with amino groups for the dyes adsorption. Although the study of these materials as adsorbents for various dyes is not a novelty, this paper is very well written, with a clear working methodology and very well explained results. Therefore, my recommendation is that the manuscript be accepted for publication.

Author Response

Reviewer 2 made no remarks.

Reviewer 3 Report

This paper deals with the preparation of Fe3O4@SiO2 particles and these properteis.

I agree with authors' statements and judge that this paper can be published in this journal.

One minor point:

For Fig. 9 and kinetic analysis, the authors proposed two models, but it agreed well with the pseudo-second order one.

Thus, the part for pseudo-first order one should be moved to supporting information or edited.

Author Response

Remark. One minor point: For Fig. 9 and kinetic analysis, the authors proposed two models, but it agreed well with the pseudo-second order one. Thus, the part for pseudo-first order one should be moved to supporting information or edited.

Answer. We would prefer to leave the description of the pseudo-first order model along with the description of the pseudo-second order one, since only after fitting the kinetic curves did it become clear that the pseudo-second order model was better suited. In the works cited by us, it is generally accepted to present the analysis of kinetics for all the models under consideration, and only then draw a conclusion about the most suitable one.